# Characterization of *Colletotrichum* Species Infecting Litchi in Hainan, China

**DOI:** 10.3390/jof9111042

**Published:** 2023-10-24

**Authors:** Xueren Cao, Fang Li, Huan Xu, Huanling Li, Shujun Wang, Guo Wang, Jonathan S. West, Jiabao Wang

**Affiliations:** 1Key Laboratory of Integrated Pest Management on Tropical Crops, Ministry of Agriculture and Rural Affairs, Hainan Key Laboratory for Monitoring and Control of Tropical Agricultural Pests, Environment and Plant Protection Institute, Chinese Academy of Tropical Agricultural Sciences, Haikou 571101, China; caoxueren1984@163.com (X.C.); wanglucai@sina.com (G.W.); 2National Key Laboratory of Green Pesticide, Key Laboratory of Green Pesticide and Agricultural Bioengineering, Ministry of Education, Center for R&D of Fine Chemicals of Guizhou University, Guiyang 550025, China; 3Rothamsted Research, Harpenden AL5 2JQ, UK; jon.west@rothamsted.ac.uk

**Keywords:** *Colletotrichum* species, litchi, identification, phylogenetic, pathogenicity

## Abstract

Litchi (*Litchi chinensis*) is an evergreen fruit tree grown in subtropical and tropical countries. China accounts for 71.5% of the total litchi cultivated area in the world. Anthracnose disease caused by *Colletotrichum* species is one of the most important diseases of litchi in China. In this study, the causal pathogens of litchi anthracnose in Hainan, China, were determined using phylogenetic and morphological analyses. The results identified eight *Colletotrichum* species from four species complexes, including a proposed new species. These were *C. karsti* from the *C. boninense* species complex; *C. gigasporum* and the proposed new species *C. danzhouense* from the *C. gigasporum* species complex; *C. arecicola*, *C. fructicola* species complex; *C. arecicola*, *C. fructicola* and *C. siamense* from the *C. gloeosporioides* species complex; and *C. musicola* and *C. plurivorum* from the *C. orchidearum* species complex. Pathogenicity tests showed that all eight species could infect litchi leaves using a wound inoculation method, although the pathogenicity was different in different species. To the best of our knowledge, the present study is the first report that identifies *C. arecicola*, *C. danzhouense*, *C. gigasporum* and *C. musicola* as etiological agents of litchi anthracnose.

## 1. Introduction

Litchi (*Litchi chinensis*), originating in southern China and possibly northern Vietnam, is an evergreen fruit tree that is now grown in subtropical and tropical countries like South Africa, Madagascar, Thailand, India and Australia. Litchi cultivation in China goes back over 2000 years and China is the largest litchi cultivation and production country, which accounts for 71.5% of the cultivated area and 62.7% of the yield in the world [1]. However, litchi quality and yield are greatly limited by plant diseases. Anthracnose, caused by *Colletotrichum* species, is one of the most important diseases of litchi in China. The disease can occur on leaves, stems, flowers and fruits [2]. The pathogens cause black to dark-brown lesions on infected tissues.

*Colletotrichum* is one of the most important genera of plant pathogenic fungi causing anthracnose on a range of economically important plant hosts [3]. Plant pathogenic *Colletotrichum* species are often described as causing typical symptoms of anthracnose disease including spots and sunken necrotic lesions on leaves, stems, flowers and fruits. Pathogen identification is the basis for plant disease monitoring and control. Traditionally, the identification of *Colletotrichum* sp. mainly relied on host range and morphological characteristics. However, these characteristics are not suitable for species identification since they are easily affected by environmental conditions [4]. Multilocus phylogenetic analyses combined with morphological data have widely been used and accepted as the basis for *Colletotrichum* species identification and many new *Colletotrichum* species have been reported [4,5,6]. In a recent study, 16 species complexes as well as 15 singleton species were classified into the genus *Colletotrichum*, and a total of 280 species are accepted in this genus [6].

Some *Colletotrichum* species from four species complexes have been reported on litchi in different countries. For example, *C. tropicale* from the *C. gloeosporioides* species complex was reported in Japan [7]; *C. queenslandicum* and *C. siamense* from the *C. gloeosporioides* species complex, *C. simmondsii* and *C. sloanei* from the *C. acutatum* species complex were reported in Australia [8]; and *C. fioriniae*, *C. guajavae* and *C. nymphaeae* from the *C. acutatum* species complex, *C. karsti* from the *C. boninense* species complex, *C. fructicola* and *C. siamense* from the *C. gloeosporioides* species complex, and *C. plurivorum* from the *C. orchidearum* species complex were reported in China [9,10,11]. These reports indicate that the *Colletotrichum* species causing diseases in litchi vary among regions.

Hainan is one of the main litchi cultivation regions in China [12]. However, only a few strains from this region were used for *Colletotrichum* species identification [9]. Therefore, more strains were obtained in this study to determine *Colletotrichum* species associated with litchi anthracnose in Hainan, China, based on phylogenetic, morphological and pathogenicity analyses.

## 2. Material and Methods

### 2.1. Sample Collection and Fungal Isolation

In 2023, litchi leaves with anthracnose symptoms were sampled from Haikou, Chengmai and Danzhou in Hainan, China. Small pieces (5 × 5 mm) of leaf tissues consisting of healthy and diseased margins were surface-sterilized with 70% ethanol for 30 s, 1% NaClO for 1 min, washed three times in sterile distilled water and dried on sterile paper. Then, the sterilized samples were placed on potato dextrose agar (PDA, 20% potato infusion, 2% dextrose, 1.5% agar and distilled water) plates and incubated at 25 °C until mycelium grew from the samples. The mycelium from the margin of the emerging mycelium was then subcultured onto new PDA plates and purified by the single-spore or single-hyphal-tip method.

Type specimens of a proposed new species herein were deposited in the Mycological Herbarium, Institute of Microbiology, Chinese Academy of Sciences, Beijing, China (HMAS). Ex-type living cultures were deposited in the China General Microbiological Culture Collection Centre (CGMCC), Beijing, China.

### 2.2. DNA Extraction, PCR Amplification and Sequencing

Fresh mycelium grown on PDA for 5 to 7 days at 25 °C was collected, and fungal genomic DNA was extracted using the Tiangen Plant Genomic DNA Kit (Tiangen Biotech, Beijing, China) with reference to the manufacturers’ protocol. Isolates were identified at the species complex level based on cultural characteristics on PDA, growth rate and partial sequences of glyceraldehyde-3-phosphate dehydrogenase (*GAPDH*). Six loci including the 5.8S nuclear ribosomal gene with the two flanking internal transcribed spacers (ITS), partial sequences of *GAPDH*, actin (*ACT*), chitin synthase 1 (*CHS-1*), beta-tubulin (*TUB2*) and the mating type locus MAT1-2 (*ApMat*), were amplified using the primer pairs ITS-1 [13]/ITS-4 [14], GDF1/GDR1 [15], ACT-512F/ACT-783R [16], CHS-79F/CHS-354R [16], T1 [17]/Bt2b [13] and AMF1/AMR1 [18], respectively.

PCR amplification was conducted in a thermal cycler (C1000; BioRad, Hercules, CA, USA). A total of 25 μL of reaction mixture including 12.5 μL of Taq-Plus PCR Forest Mix (NOVA, Lianyungang, China), 1 μL of DNA template, 1 μL of each primer (5 μM) and 9.5 μL of ddH_2_O was used. PCR reactions for *GAPDH* were performed using the following conditions: initial denaturation at 95 °C for 4 min, followed by 35 cycles each consisting of 30 s at 95 °C, 30 s at 60 °C plus an extension for 45 s at 72 °C, with a final extension step at 72 °C for 7 min. PCR conditions for the other five loci were the same as for *GAPDH* except the annealing temperatures: ITS at 52 °C, *ACT* at 58 °C, *TUB2* at 55 °C, *CHS-1* at 58 °C and *ApMat* at 62 °C.

PCR products were examined by electrophoresis in 1.0% agarose gels stained with GoodView Nucleic Acid Stain (Beijing SBS Genetech, Beijing, China) and photographed under UV light. The PCR products were sent to the Sangon Biotech Company, Ltd. (Shanghai, China) for DNA purifying and sequencing. Consensus sequences were obtained by assembling the forward and reverse sequences with DNAMAN (v. 9.0; Lynnon Bio soft). Sequences generated in the current study were submitted to GenBank and the accession numbers are listed in Table 1.

### 2.3. Phylogenetic Analyses

Isolates were divided into two groups for multilocus phylogenetic analyses, and type isolates of each species were selected and included in the analyses (Table 1). Multiple sequence alignments of each locus were prepared using ClustalW (implemented in MEGA 6.0) and manually edited if necessary. Bayesian inference (BI) was used to construct phylogenies using MrBayes v. 3.2.6 [19]. The optimal nucleotide substitution model for each locus was determined using MrModeltest v. 2.3 [20] based on the corrected Akaike information criterion (AIC). For the *C. gloeosporioides* species complex, the following nucleotide substitution models were used: SYM + I + G for ITS, HKY + I + G for *GAPDH*, K80 + G for *CHS-1*, GTR + G for *ACT* and *TUB2*, and HKY + G for *ApMat*, and they were all incorporated in the analysis. For the isolates from the other three species complexes, the following models were used: SYM + I + G for ITS, HKY + I + G for *GAPDH*, *CHS-1* and *TUB2* and GTR + I + G for *ACT*, and they were all incorporated in the analysis. Two analyses of four Markov chain Monte Carlo (MCMC) chains were run from random trees with 4 × 10^6^ generations for the *C. gloeosporioides* species complex and 2 × 10^6^ for other three *Colletotrichum* species complexes. The analyses were sampled every 1000 generations and stopped when standard deviation of split frequencies fell below 0.01. The first 25% of trees were discarded as the burn-in phase of each analysis and posterior probability values were calculated. Clades were regarded as significantly supported if they had a posterior probability ≥0.95 [19]. Furthermore, maximum likelihood (ML) analyses of the multilocus alignments were conducted using RaxmlGUI v. 1.3.1 [21] using a GTRGAMMAI substitution model with 1000 bootstrap replicates. The phylogenetic trees constructed in this study were submitted to TreeBASE (accession number: S30748).

New species and their most closely related neighbors were analyzed using the Genealogical Concordance Phylogenetic Species Recognition (GCPSR) model by performing a pairwise homoplasy index (PHI) test [22]. The PHI test was performed in SplitsTree 4.14.5 [23,24] using concatenated sequences (ITS, *GAPDH*, *ACT*, *CHS-1*, and *TUB2*) to determine the recombination level within phylogenetically closely related species. The relationship between closely related species was visualized by constructing a split graph.

### 2.4. Morphological Analysis

The species identified by phylogenetic analysis were selected for morphological characterization. Fresh mycelial discs (5 mm diameter), cut from the edge of 5-day-old colonies, were transferred to new PDA and cultivated at 25 °C in the dark. After 7 day, the colony characteristics were recorded, and colony diameters were measured to calculate fungal growth rate. The conidia shape and size were observed using a light microscope (Eclipse 80i, Nikon, Tokyo, Japan) (30 conidia were selected randomly for each strain). For the new proposed species, morphological and cultural features on oatmeal agar (OM) and synthetic nutrient-poor agar medium (SNA) were also studied.

### 2.5. Pathogenicity Tests

Young healthy leaves of litchi (cv. Feizixiao), the most widely planted litchi species in China [25], were collected for pathogenicity tests using both wound and nonwound inoculation methods. The tested leaves were washed three times in sterile water and then air-dried on sterilized papers. The left side of the midrib of each leaf was wounded with a sterilized needle (insect pin, 0.5 mm diameter) and then 6 μL of conidial suspension (10^6^ conidia per mL) was dropped on the wound of the left side of the leaf. Similarly, conidial suspension was dropped on the right side of the same leaf without wounding. Three replicates were used for each isolate and each replicate consisted of two leaves. Leaves inoculated with sterile water onto the wound or nonwound was considered as the controls. Treated leaves were put on moist tissue paper in plastic trays, maintained in a moist chamber at 25 °C with a 12 h day/night regime and monitored daily for lesion development. The lesion diameter was measured 4 days after inoculation. The experiment was performed twice. The fungus was reisolated from the resulting lesions and identified as described above, thus fulfilling Koch’s postulates.

## 3. Results

### 3.1. Colletotrichum Isolates Associated with Litchi Anthracnose

A total of 61 *Colletotrichum* isolates were obtained based on morphology and *GAPDH* sequence data. Based on the BLAST results of the *GAPDH* sequences, the 61 *Colletotrichum* isolates were from four species complexes, including the *C. boninense* species complex (one isolates), *C. gigasporum* species complex (six isolates), *C. gloeosporioides* species complex (forty-eight isolates) and *C. orchidearum* species complex (six isolates). A total of thirty-eight representative isolates (one, five, twenty-eight and four isolates from the *C. boninense*, *C. gigasporum*, *C. gloeosporioides* and *C. orchidearum* species complex, respectively) were chosen for further species identification based on their morphology (colony characters), *GAPDH* sequence data and origin (Table 1).

### 3.2. Multilocus Phylogenetic Analyses

A multilocus phylogenetic analysis with the concatenated ITS, *GAPDH*, *CHS-1*, *ACT*, *TUB2* and *ApMat* sequences was carried out for the isolates from the *C. gloeosporioides* species complex including 28 isolates from litchi in this study, 51 reference isolates from the *C. gloeosporioides* species complex and the outgroup *C. boninense* (ICMP 17904) (Figure 1). The combined gene alignment contained 3200 characters including gaps (gene/locus boundaries of ITS: 1–617, *GAPDH*: 618–927, *CHS-1*: 928–1226, *ACT*: 1227–1534, *TUB2*: 1535–2261, *ApMat*: 2262–3200) and the Bayesian analysis was performed based on 1445 unique site patterns (ITS: 142, *GAPDH*: 224, *CHS-1*: 85, *ACT*: 140, *TUB2*: 291, *ApMat*: 563). The maximum likelihood tree confirmed the tree topology from the Bayesian analysis. As the phylogenetic tree shows in Figure 1, for the 28 isolates in the *C. gloeosporioides* species complex, 1 isolate was clustered with *C. arecicola* (Bayesian posterior probabilities value 1/RAxML bootstrap support value 99), 5 with *C. fructicola* (0.95/95) and 22 with *C. siamense* (1/68) (Figure 1).

For isolates belonging to the other species complexes, the alignment of combined DNA sequences was obtained from 63 taxa, including 10 isolates from litchi in this study, 52 reference isolates of *Colletotrichum* species, and 1 outgroup strain *C. gloeosporioides* (ICMP 17821) (Figure 2). The gene/locus boundaries of the aligned 2299 characters (with gaps) were ITS: 1–618, *GAPDH*: 619–946, *CHS-1*: 947–1245, *ACT*: 1246–1539 and *TUB2*: 1540–2299), and the Bayesian analysis was performed based on 1169 unique site patterns (ITS: 197, *GAPDH*: 283, *CHS-1*: 117, *ACT*: 170, *TUB2*: 402). The maximum likelihood tree confirmed the tree topology from the Bayesian analysis. For the four isolates in the *C. orchidearum* species complex, one isolate was grouped with *C. musicola* (1/84) and three with *C. plurivorum* (1/96). One isolate from the *C. boninense* species complex was identified as *C. karsti* (1/99). For the five isolates in the *C. gigasporum* species complex, three of them were grouped with *C. gigasporum* (1/100), while the other two formed a clade distantly from any reported species in this complex, which was described as a new species, *C. danzhouense*, in this study (Figure 2). The application of the PHI test to the concatenated five-locus sequences (ITS, *GAPDH*, *ACT*, *CHS-1* and TUB2) revealed that no significant recombination event (*p* = 0.14) occurred between *C. danzhouense* and phylogenetically related species *C. gigasporum* and *C. zhaoqingense* (Figure 3). This is further evidence that *C. danzhouense* is a new species.


**Taxonomy**


*Colletotrichum danzhouense* Fungal Names Number: FN 571654; Figure 4.

Etymology: Named after the location where the fungus was sampled, Danzhou city.

Type: China, Hainan province, Danzhou City, from diseased leaves of *Litchi chinensis*, 15 May 2023, X. R. Cao, holotype (HMAS 352507), ex-type living culture CGMCC 3.25375 = DL 52.

*Description*: Sexual morph not observed. *Vegetative hyphae* septate, hyaline, smooth-walled, branched. Conidia and setae not observed on PDA or OA. On SNA, *conidiomata* acervular, scattered, in which conidiophores are hardly observed. *Setae* 1–4 septate, 70.8–113.4 μm long, basal cells cylindrical, smooth-walled, 4.1–6.8 μm diameter, tip acute. *Conidiophores*, formed directly on hyphae, usually reduced to conidiogenous cells. *Conidiogenous cells* hyaline, cylindrical, formed terminally or laterally on hyphae, variable in size. *Conidia* hyaline, cylindrical with obtuse ends, smooth-walled, granular, 14.4–21.6 × 5.6–7.2 μm, mean ± SD = 17.6 ± 1.7 × 6.5 ± 0.4 μm, L/W ratio = 2.7. *Appressoria* variable in shape, pale brown, 9.7–19.2 × 8.5–14.3 μm, mean ± SD = 13.2 ± 2.1 × 10.5 ± 1.6 μm, L/W ratio = 1.3.

Culture characteristics: *Colonies* on PDA flat with entire edge, gray to pale green with a white margin, aerial mycelium floccose, reverse dark green in the center with a white margin. Colonies’ diameters of 52–54 mm, 80–85 mm and 40–44 mm in 7 day incubated at 25 °C on PDA, SNA and OA, respectively. Conidia and setae not observed on PDA or OA.

Additional specimens examined: China, Hainan province, Danzhou City, from diseased leaves of *Litchi chinensis*, 15 May 2023, X. R. Cao, living culture DL 107.

Notes: *Colletotrichum danzhouense* is phylogenetically closely related to *C. gigasporum* and *C. zhaoqingense* in the *C. gigasporum* species complex (Figure 2); it was isolated from infected litchi leaves collected from Danzhou in Hainan, China. It shares a low sequence similarity with *C. gigasporum* at *GAPDH* (89.5%), *CHS-1* (96.0%) and TUB2 (96.4%). Also, a low sequence similarity was observed between the new species and *C. zhaoqingense* at *GAPDH* (89.8%), *CHS-1* (96.3%) and TUB2 (96.1%). In morphology, *C. danzhouense* differs from *C. gigasporum* and *C. zhaoqingense* by producing shorter conidia (14.4–21.6 × 5.6–7.2 μm vs. 22–32 × 6–9 μm, 14.4–21.6 × 5.6–7.2 μm vs. 20–24 × 5.5–7 μm, respectively).

### 3.3. Morphological and Cultural Characterization

All species produced dense mycelium except *C. karstii* (Table 2). *C. gigasporum* produced larger conidia compared with other species identified in the present study. The three species, *C. arecicola*, *C. fructicola* and *C. siamense*, from the *C. gloeosporioides* species complex had similar conidia size, while the conidia size was different between the two species, *C. danzhouense* and *C. gigasporum*, from the *C. gigasporum* species complex. Additionally, the width of the conidia from these three species was smaller than that of the other five species obtained in this study. The L/W ratio of the conidia of *C. karstii* was smaller, while *C. gigasporum* had a larger L/W ratio. The growth rates of *C. danzhouense*, *C. karstii* and *C. musicola* were relatively slow at <9 mm/d while the growth rate was higher than 11 mm/d for the other five *Colletotrichum* species obtained in this study (Table 2).

### 3.4. Pathogenicity Tests

Eight *Colletotrichum* species were able to infect litchi leaves (cv. Feizixiao) and cause typical symptoms of anthracnose when inoculated onto wounded leaves (Figure 5) with an average lesion diameter ranging from 2.3 to 9.7 mm 4 days after inoculation (Figure 6). The diameters of lesions for *C. fructicola* and *C. siamense* (>9 mm) from the *C. gloeosporioides* species complex were significantly larger than those produced by other species except *C. arecicola*. The proposed new species, *C. danzhouense*, produced significantly larger lesion (>5.5 mm) than *C. musicola*, *C. plurivorum* and *C. karstii*, while the diameter of *C. karstii* was the smallest. However, five of the eight species did not produce visible symptoms on litchi leaves when nonwounded sites were inoculated, whereas *C. danzhouense*, *C. fructicola* and *C. siamense* did produce lesions on nonwounded, inoculated leaves (Figure 5).

## 4. Discussion

In this study, pathogens from four *Colletotrichum* species complexes were found to cause litchi anthracnose in Hainan, China, and *C. gigasporum* species complex was first reported to cause anthracnose on litchi based on morphological and multilocus sequences. Nearly 80% of the isolates obtained in the present study belonged to the *C. gloeosporioides* species complex, which was consistent with previous reports that *C. gloeosporioides* was the main pathogen of litchi anthracnose [26,27].

A total of eight *Colletotrichum* species were found to be responsible for anthracnose of litchi in Hainan, China. Three of them (*C. arecicola*, *C. fructicola* and *C. siamense*) were from the *C. gloeosporioides* species complex. The former two species were reported on litchi [8,9]. *C. siamense* was the most common species to cause anthracnose of litchi in Hainan in this study. Also, this species was the dominant species associated with anthracnose of rubber tree, coffee and areca palm in Hainan [28,29,30]. Both rubber tree and areca palm are widely cultivated in Hainan, which is a likely factor contributing to the pathogen cross-infecting other hosts. *C. fructicola* is a plant pathogen with a broad host range [6]. Also, this species was reported on rubber tree, coffee and areca palm in Hainan. Furthermore, it was proved to be the most predominant species causing tea-oil camellia anthracnose in Hainan [31]. In this study, *C. fructicola* was isolated from litchi. *C. arecicola*, which had previously been reported only on areca palm in Hainan [30], was found on litchi for the first time in the present study.

*Colletotrichum karsti* from the *C. boninense* species complex is another species commonly detected in China with a broad host range [6]. This species was also reported on litchi in Guangxi, China [11]. In this study, *C*. *karsti* was also obtained on litchi in Hainan although only one isolate was obtained.

Two species from the *C. orchidearum* species complex were isolated in this study. One of them, *C. plurivorum*, has a broad host range and has been reported on litchi before [9]. The other species was *C. musicola*, which was first reported on *Musa* sp. [32]. Then, this species was reported on *Colocasia esculenta* [33], *Glycine max* [34] and *Manihot esculenta* [35]. This study is the first to demonstrate that this species can also occur on litchi, although it was found with a low frequency.

*Colletotrichum gigasporum* from the *C. gigasporum* species complex was reported as a causal agent of anthracnose disease on coffee and mango in Hainan [29,36]. This is the first report of this species on litchi. Furthermore, *C. danzhouense*, which clustered with *C. gigasporum* and *C. zhaoqingense*, was proposed as a new species in the *C. gigasporum* species complex because it had a low sequence similarity to the other two species at *GAPDH*, *CHS-1* and TUB2. The BLAST results of the *GAPDH* and ITS sequences indicated that this species was most similar to *Colletotrichum* sp. Also, no significant recombination event (*p* = 0.14) occurred among these three species. Furthermore, *C. danzhouense* produced shorter conidia compared with *C. gigasporum* and *C. zhaoqingense*.

*Colletotrichum* species from the *C. acutatum* species complex was also reported previously as the pathogen causing litchi anthracnose in Australia and China [8,9]. However, no isolates from this complex were obtained in this study. The main reason may be the geographic distribution of the pathogen and the different sample sites studied. Also, *C. acutatum* was only occasionally obtained from litchi in a previous study [26], but it was rarely found.

Wounding is known to enhance *Colletotrichum* infection and disease development. Furthermore, for grape leaf [37] and mango fruit [38], wounding is necessary for *Colletotrichum* to infect. Only 2 of 12 *Colletotrichum* species from cultivated pear were pathogenic to pear leaves inoculated without wounding [39]. This was also observed in this study, as the pathogenicity tests indicated that all eight species isolated were able to infect litchi leaves when inoculated onto wounded leaves, while only three of the eight species induced visible symptoms on litchi leaves using a nonwound inoculation method. One reason could be that the cuticle and epidermis may act as a barrier for the infection by *Colletotrichum* spp. [40]. Alternatively, the quiescent infection, which means that the infection of healthy intact leaves may produce visual symptoms only at a later stage when the leaf physiological state changes significantly, is an important feature of *Colletotrichum* spp. [41]. In field conditions, wounds on litchi leaves can be common in nature due to wind, insect damage and abrasions caused by leaves rubbing. Generally, isolates from *C. fructicola* and *C. siamense* were found to cause larger lesions than those caused by other species. These two species were also the most common species obtained in this study.

In conclusion, eight *Colletotrichum* species from four species complexes were demonstrated as pathogens causing litchi anthracnose in Hainan, China; one species complex and four species were reported on litchi for the first time. The results of this study can be valuable for developing sustainable management strategies for anthracnose of litchi. The precise identification of fungal pathogens is important for disease control measures. Currently, the main strategy for litchi anthracnose management is fungicide application [2]. It was reported that *Colletotrichum* species displayed differential sensitivity to fungicides [29,42]. Therefore, it is essential to determine the species in a given plantation before fungicide applications.

## Figures and Tables

**Figure 1 jof-09-01042-f001:**
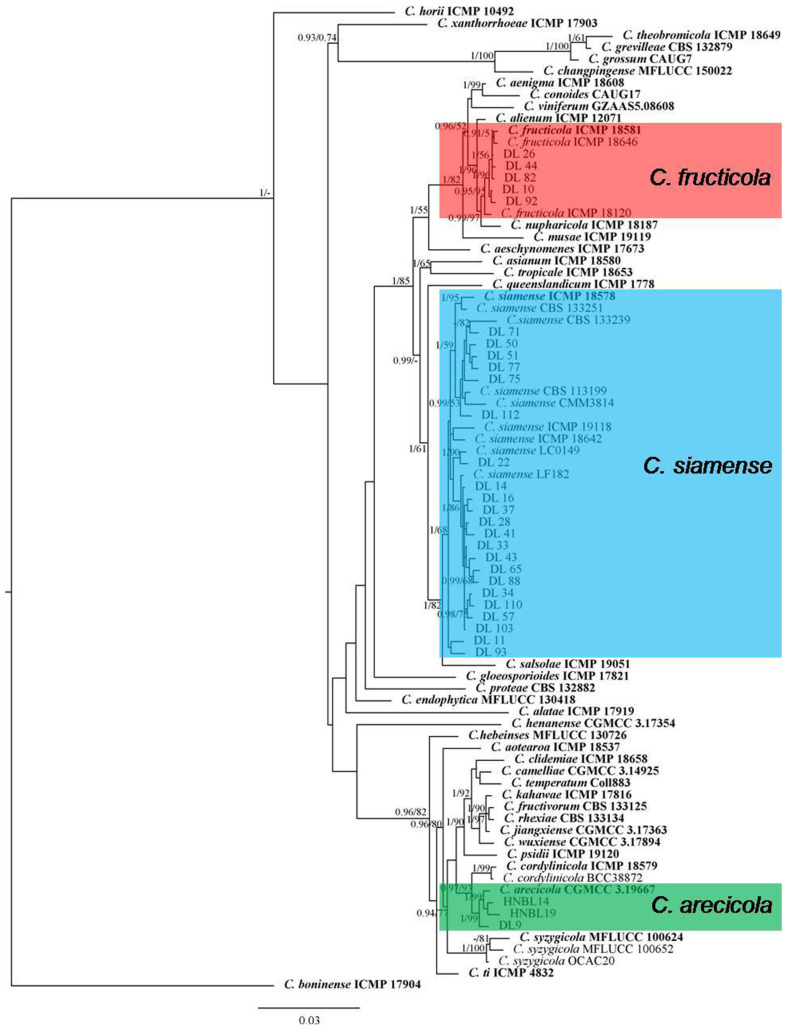
A Bayesian inference phylogenetic tree built using concatenated sequences of ITS, *ACT*, *CHS-1*, *GAPDH*, *TUB2* and *ApMat* for the *Colletotrichum* spp. isolates from the *C. gloeosporioides* species complex. The species *C. boninense* (ICMP 17904) was used as an outgroup. Bayesian posterior probability values (PP ≥ 0.90) and RAxML bootstrap support values (ML ≥ 50%) are shown at the nodes. Ex-type isolates are shown in **bold**. Colored blocks indicate clades including isolates obtained in this study.

**Figure 2 jof-09-01042-f002:**
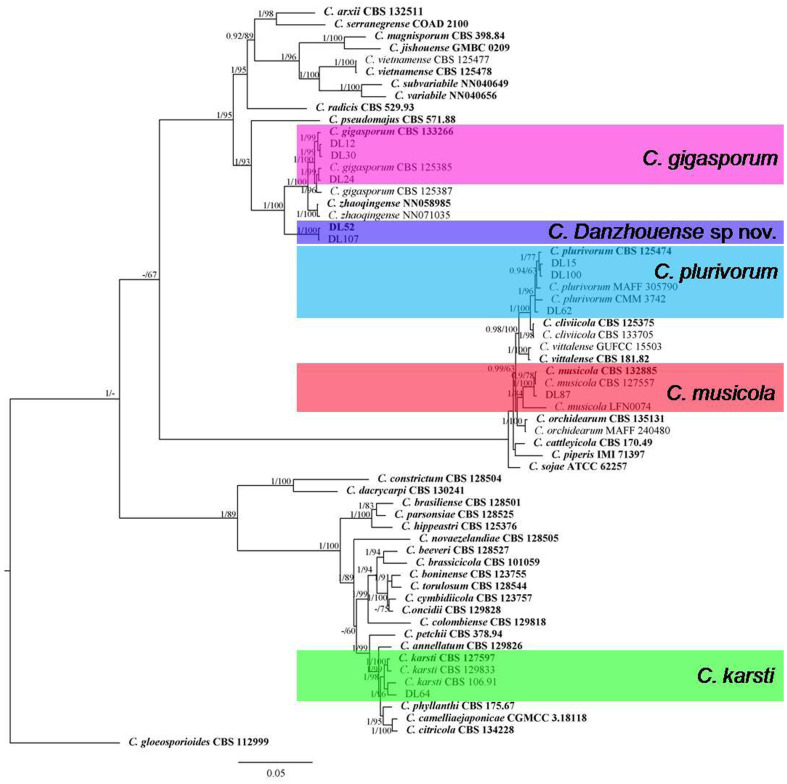
A Bayesian inference phylogenetic tree built using concatenated sequences of ITS, *ACT*, *CHS-1*, *GAPDH* and *TUB2* for the *Colletotrichum* spp. isolates from the *C. gigasporum*, *C. orchidearum* and *C. boninense* species complex with *C. gloeosporioides* (ICMP 17821) as an outgroup. Bayesian posterior probability values (PP ≥ 0.90) and RAxML bootstrap support values (ML ≥ 50%) are shown at the nodes. Ex-type isolates are shown in **bold**. Colored blocks indicate clades including isolates obtained in this study.

**Figure 3 jof-09-01042-f003:**
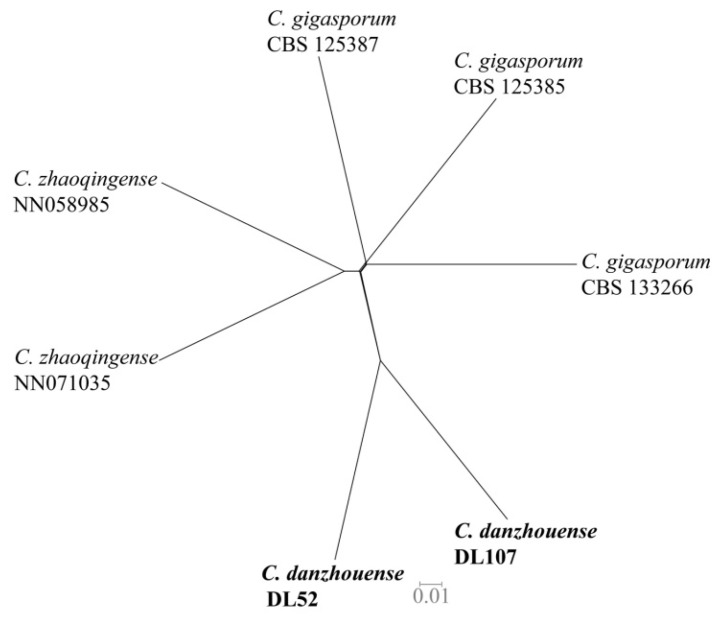
The result of the pairwise homoplasy index (PHI) of *Colletotrichum danzhouense* and its phylogenetically related species using both a LogDet transformation and splits decomposition. No significant recombination event (*p* = 0.14) was observed within the datasets. Isolates obtained in this study are shown in **bold**.

**Figure 4 jof-09-01042-f004:**
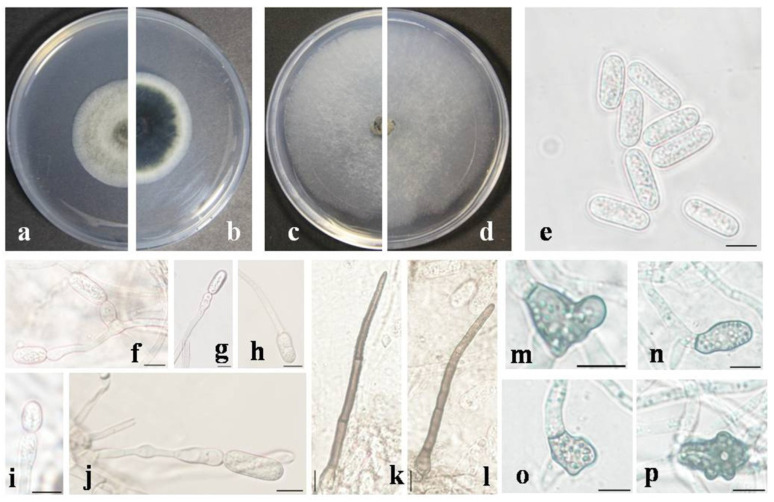
Morphological characteristics of *Colletotrichum danzhouense*. (**a**,**b**) Front and reverse colony on PDA (7 day); (**c**,**d**) front and reverse colony on SNA (7 day); (**e**) conidia; (**f**–**j**) conidiophores; (**k**,**l**) setae; (**m**–**p**) appressoria; (**e**–**p**) produced on SNA. Scale bars = 10 μm.

**Figure 5 jof-09-01042-f005:**
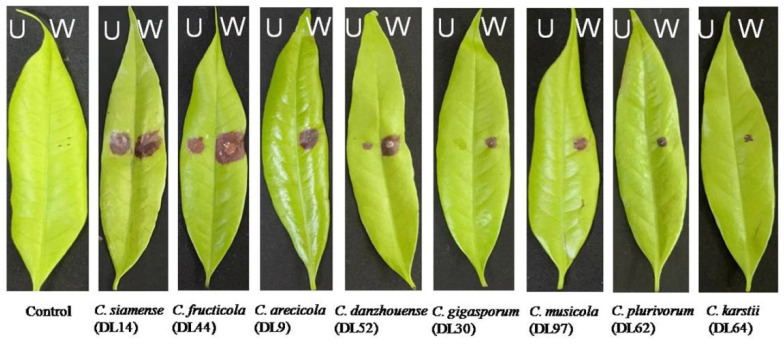
Symptoms of litchi leaves (cv. Feizixiao) induced by inoculation of spore suspensions of eight *Colletotrichum* spp. after four days at 25 °C under unwounded (U) and wounded (W) conditions.

**Figure 6 jof-09-01042-f006:**
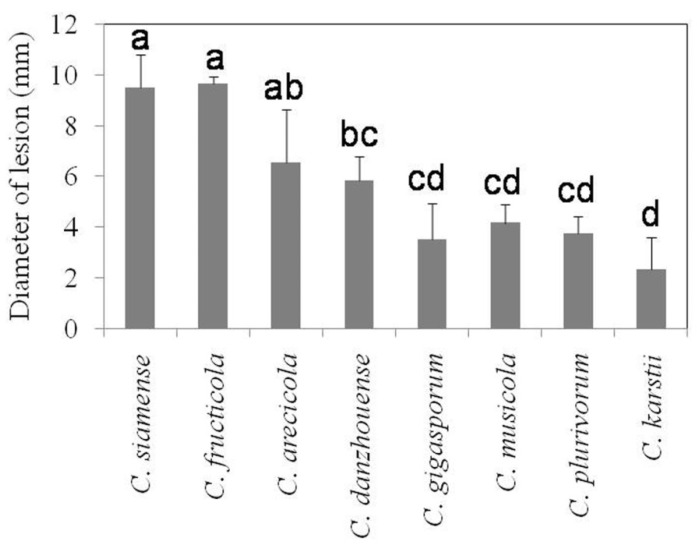
Lesion diameters of *Colletotrichum* species on litchi leaves (cv. Feizixiao) using wound inoculation methods. Letters over the error bars indicate a significant difference at the *p* = 0.05 level.

**Table 1 jof-09-01042-t001:** Strains of the *Colletotrichum* species with details of host, location and GenBank accessions of the sequences.

Taxon	Isolate Designation	Host	Location	ITS	Gapdh	Act	chs1	tub2	ApMat
*C. aenigma*	ICMP 18608 *	*Persea americana*	Israel	JX010244	JX010044	JX009443	JX009774	JX010389	KM360143
*C. aeschynomenes*	ICMP17673 *	*Aeschynomene virginica*	USA	JX010176	JX009930	JX009483	JX009799	JX010392	KM360145
*C. alatae*	ICMP 17919 *	*Dioscorea alata*	India	JX010190	JX009990	JX009471	JX009837	JX010383	KC888932
*C. alienum*	ICMP12071	*Malus domestica*	New Zealand	JX010251	JX010028	JX009572	JX009882	JX010411	KM360144
*C. annellatum*	CBS 129826, CH1 *	*Hevea brasiliensis*	Colombia	JQ005222	JQ005309	JQ005570	JQ005396	JQ005656	-
*C. aotearoa*	ICMP18537	*Coprosma* sp.	New Zealand	JX010205	JX010005	JX009564	JX009853	JX010420	KC888930
*C. arecicola*	CGMCC 3.19667, HNBL5 *	*Areca catechu*	China	MK914635	MK935455	MK935374	MK935541	MK935498	MK935413
	HNBL14	*Areca catechu*	China	MK914641	MK935461	MK935380	MK935547	MK935504	MK935419
	HNBL19	*Areca catechu*	China	MK914643	MK935463	MK935382	MK935549	MK935506	MK935421
	**DL9**	** *Litchi chinensis* **	**China**	**OR461235**	**OR455971**	**OR456047**	**OR456009**	**OR456085**	**OR456113**
*C. arxii*	CBS 132511, Paphi 2-1 *	*Paphiopedilum* sp.	Germany	KF687716	KF687843	KF687802	KF687780	KF687881	-
*C. asianum*	ICMP 18580, CBS 130418 *	*Coffea arabica*	Thailand	FJ972612	JX010053	JX009584	JX009867	JX010406	FR718814
*C. cattleyicola*	CBS 170.49 *	*Cattleya* sp.	Belgium	MG600758	MG600819	MG600963	MG600866	MG601025	
*C. beeveri*	CBS 128527, ICMP 18594 *	*Brachyglottis repanda*	New Zealand	JQ005171	JQ005258	JQ005519	JQ005345	JQ005605	-
*C. boninense*	ICMP 17904, CBS 123755, MAFF 305972 *	*Crinum asiaticum* var. *sinicum*	Japan	JQ005153	JQ005240	JQ005501	JQ005327	JQ005588	-
*C. brasiliense*	CBS 128501, ICMP 18607 *	*Passiflora edulis*	Brazil	JQ005235	JQ005322	JQ005583	JQ005409	JQ005669	-
*C. brassicicola*	CBS 101059, LYN 16331 *	*Brassica oleracea* var. *gemmifera*	New Zealand	JQ005172	JQ005259	JQ005520	JQ005346	JQ005606	-
*C. camelliae*	CGMCC 3.14925 *	*Camellia sinensis*	China	KJ955081	KJ954782	KJ954363	-	KJ955230	KJ954497
*C. camelliae-japonicae*	CGMCC3.18118 *	*Camellia japonica*	Japan	KX853165	KX893584	KX893576	-	KX893580	-
*C. changpingense*	MFLUCC 15-0022 *	*Fragaria× ananassa*	China	KP683152	KP852469	KP683093	KP852449	KP852490	-
*C. clidemiae*	ICMP18658 *	*Clidemia hirta*	USA	JX010265	JX009989	JX009537	JX009877	JX010438	KC888929
*C. cliviicola*	CBS 125375 *	*Clivia miniata*	China	MG600733	MG600795	MG600939	MG600850	MG601000	-
	CBS 133705	*Clivia* sp.	South Africa	MG600732	MG600794	MG600938	MG600849	MG600999	-
*C. citricola*	CBS 134228,SXC151 ***	*Citrus unchiu*	China	KC293576	KC293736	KC293616	KC293792	KC293656	-
*C. colombiense*	CBS 129818, G2 *	*Passiflora edulis*	Colombia	JQ005174	JQ005261	JQ005522	JQ005348	JQ005608	-
*C. conoides*	CGMCC 3.17615, CAUG17 *	*Capsicum annuum*	China	KP890168	KP890162	KP890144	KP890156	KP890174	-
*C. constrictum*	CBS 128504, ICMP 12941 *	*Citrus limon*	New Zealand	JQ005238	JQ005325	JQ005586	JQ005412	JQ005672	-
*C. cordylinicola*	ICMP 18579, MFLUCC 090551 *	*Cordyline fruticosa*	Thailand	JX010226	JX009975	HM470233	JX009864	JX010440	JQ899274
	BCC38872	*Codyline fruticosa*	-	HM470246	HM470240	HM470234	-	HM47029	-
*C. cymbidiicola*	CBS 123757, MAFF 306100 *	*Cymbidium* sp.	Japan	JQ005168	JQ005255	JQ005516	JQ005342	JQ005602	-
*C. dacrycarpi*	CBS 130241, ICMP 19107 *	*Dacrycarpus dacrydioides*	New Zealand	JQ005236	JQ005323	JQ005584	JQ005410	JQ005670	-
** *C. danzhouense* **	**CGMCC 3.25375, DL52 ***	** *Litchi chinensis* **	**China**	**OR461229**	**OR455965**	**OR456041**	**OR456003**	**OR456079**	-
	**DL107**	** *Litchi chinensis* **	**China**	**OR461230**	**OR455966**	**OR456042**	**OR456004**	**OR456080**	-
*C. endophytica*	MFLUCC 13–0418 *	*Pennisetum purpureum*	Thailand	KC633854	KC832854	KF306258	-	-	-
*C. fructicola*	ICMP18581, CBS 130416 *	*Coffea arabica*	Thailand	JX010165	JX010033	FJ907426	JX009866	JX010405	JQ807838
	ICMP 18646, CBS 125397	*Tetragastris panamensis*	Panama	JX010173	JX010032	JX009581	JX009874	JX010409	JQ807839
	ICMP 18120	*Dioscorea alata*	Nigeria	JX010182	JX010041	JX009436	JX009844	JX010401	-
	**DL10**	** *Litchi chinensis* **	**China**	**OR461236**	**OR455972**	**OR456048**	**OR456010**	**OR456086**	**OR456114**
	**DL26**	** *Litchi chinensis* **	**China**	**OR461237**	**OR455973**	**OR456049**	**OR456011**	**OR456087**	**OR456115**
	**DL44**	** *Litchi chinensis* **	**China**	**OR461238**	**OR455974**	**OR456050**	**OR456012**	**OR456088**	**OR456116**
	**DL82**	** *Litchi chinensis* **	**China**	**OR461239**	**OR455975**	**OR456051**	**OR456013**	**OR456089**	**OR456117**
	**DL92**	** *Litchi chinensis* **	**China**	**OR461240**	**OR455976**	**OR456052**	**OR456014**	**OR456090**	**OR456118**
*C. fructivorum*	Coll1414,CBS 133125 *	*Vaccinium macrocarpon*	USA	JX145145	-	-	-	JX145196	JX145300
*C. gigasporum*	CBS 133266, MuCL 44947 *	*Centella asiatica*	Madagascar	KF687715	KF687822	-	KF687761	KF687866	-
	CBS 125385, E2452	*Virola surinamensis*	Panama	KF687732	KF687835	KF687787	KF687764	KF687872	-
	CBS 125387, 4801	*Theobroma cacao*	Panama	KF687733	KF687834	KF687765	KF687788	KF687873	-
	**DL12**	** *Litchi chinensis* **	**China**	**OR461226**	**OR455962**	**OR456038**	**OR456000**	**OR456076**	-
	**DL24**	** *Litchi chinensis* **	**China**	**OR461227**	**OR455963**	**OR456039**	**OR456001**	**OR456077**	-
	**DL30**	** *Litchi chinensis* **	**China**	**OR461228**	**OR455964**	**OR456040**	**OR456002**	**OR456078**	-
*C. gloeosporioides*	ICMP 17821,CBS 112999, IMI 356878 *	*Citrus sinensis*	Italy	JX010152	JX010056	JX009531	JX009818	JX010445	JQ807843
*C. grevilleae*	CBS 132879 *	*Grevillea* sp.	Italy	KC297078	KC297010	KC296941	KC296987	KC297102	-
*C. grossum*	CAUG7, CGMCC3.17614 *	*Capsicum* sp.	China	KP890165	KP890159	KP890141	KP890153	KP890171	-
*C. hebeinses*	MFLUCC13–0726 *	*Vitis vinifera* cv. Cabernet Sauvignon	China	KF156863	KF377495	KF377532	KF289008	KF288975	-
*C. henanense*	CGMCC 3.17354 *	*Camellia sinensis*	China	KJ955109	KJ954810	KM023257	-	KJ955257	KJ954524
*C. hippeastri*	CBS 125376, CSSG1 *	*Hippeastrum vittatum*	China	JQ005231	JQ005318	JQ005579	JQ005405	JQ005665	-
*C. horii*	ICMP 10492, NBRC 7478 *	*Diospyros kaki*	Japan	GQ329690	GQ329681	JX009438	JX009752	JX010450	JQ807840
*C. jiangxiense*	CGMCC 3.17363 *	*Camellia sinensis*	China	KJ955201	KJ954902	KJ954471	-	KJ955348	KJ954607
*C. jishouense*	GZU HJ2 G3, GMBC 0209 *	*Nothapodytes pittosporoides*	China	MH482929	MH681658	MH708135	-	MH727473	-
*C. kahawae*	ICMP 17816, IMI 319418 *	*Coffea arabica*	Kenya	JX010231	JX010012	JX009452	JX009813	JX010444	JQ899282
*C. karstii*	CBS 127597, BRIP 29085a *	*Diospyros australis*	Australia	JQ005204	JQ005291	JQ005552	JQ005378	JQ005638	-
	CBS 129833	*Musa* sp.	Mexico	JQ005175	JQ005262	JQ005523	JQ005349	JQ005609	-
	CBS 106.91	*Carica papaya*	Brazil	JQ005220	JQ005307	JQ005568	JQ005394	JQ005654	-
	**DL64**	** *Litchi chinensis* **	**China**	**OR461225**	**OR455961**	**OR456037**	**OR455999**	**OR456075**	-
*C. magnisporum*	CBS 398.84 *	unknown	unknown	KF687718	KF687842	KF687803	KF687782	KF687882	-
*C. musae*	ICMP 19119, CBS 116870 *	*Musa* sp.	USA	JX010146	JX010050	JX009433	JX009896	HQ596280	KC888926
*C. musicola*	CBS 132885 *	*Musa* sp.	Mexico	MG600736	MG600798	MG600942	MG600853	MG601003	-
	CBS 127557	*Musa* sp.	Mexico	MG600737	MG600799	MG600943	MG600854	MG601004	-
	LFN0074	*Colocasia esculenta*	Mexico	MK882586	MK882587	MK882587	-	MK142675	-
	**DL87**	** *Litchi chinensis* **	**China**	**OR461234**	**OR455970**	**OR456046**	**OR456008**	**OR456084**	-
*C. novae-zelandiae*	ICMP 12944, CBS 128505 *	*Capsicum annuum*	New Zealand	JQ005228	JQ005315	JQ005576	JQ005402	JQ005662	-
*C. nupharicola*	ICMP 18187 *	*Nuphar lutea* subsp. *polysepala*	USA	JX010187	JX009972	JX009437	JX009835	JX010398	JX145319
*C. oncidii*	CBS 129828 *	*Oncidium* sp.	Germany	JQ005169	JQ005256	JQ005517	JQ005343	JQ005603	-
*C. orchidearum*	CBS 135131 *	*Dendrobium nobile*	Netherlands	MG600738	MG600800	MG600944	MG600855	MG601005	-
	MAFF 240480	*Dendrobium phalaenopsis*	Japan	MG600746	MG600808	MG600952	MG600858	MG601013	-
*C. parsonsiae*	CBS 128525, ICMP 18590 *	*Parsonsia capsularis*	New Zealand	JQ005233	JQ005320	JQ005581	JQ005407	JQ005667	-
*C. petchii*	CBS 378.94 *	*Dracaena marginata*	Italy	JQ005223	JQ005310	JQ005571	JQ005397	JQ005657	-
*C. piperis*	IMI 71397,CPC 21195 *	*Piper nigrum*	Malaysia	MG600760	MG600820	MG600964	MG600867	MG601027	-
*C. phyllanthi*	CBS 175.67, MACS 271 *	*Phyllanthus acidus*	India	JQ005221	JQ005308	JQ005569	JQ005395	JQ005655	-
*C. plurivorum*	CBS 125474 *	*Coffea* sp.	Vietnam	MG600718	MG600781	MG600925	MG600841	MG600985	-
	CMM 3742	*Mangifera indica*	Brazil	KC702980	KC702941	KC702908	KC598100	KC992327	-
	MAFF 305790	*Musa* sp.	Japan	MG600726	MG600789	MG600932	MG600845	MG600993	-
	**DL15**	** *Litchi chinensis* **	**China**	**OR461231**	**OR455967**	**OR456043**	**OR456005**	**OR456081**	-
	**DL62**	** *Litchi chinensis* **	**China**	**OR461232**	**OR455968**	**OR456044**	**OR456006**	**OR456082**	-
	**DL100**	** *Litchi chinensis* **	**China**	**OR461233**	**OR455969**	**OR456045**	**OR456007**	**OR456083**	-
*C. proteae*	CBS 132882 *	*Proteaceae*	South Africa	KC297079	KC297009	KC296940	KC296986	KC297101	-
*C. pseudomajus*	CBS 571.88 *	*Camellia sinensis*	Taiwan	KF687722	KF687826	KF687801	KF687779	KF687883	-
*C. psidii*	ICMP 19120 *	*Psidium* sp.	Italy	JX010219	JX009967	JX009515	JX009901	JX010443	KC888931
*C. queenslandicum*	ICMP 1778 *	*Carica papaya*	Australia	JX010276	JX009934	JX009447	JX009899	JX010414	KC888928
*C. radicis*	CBS 529.93 *	unknown	Costa Rica	KF687719	KF687825	KF687785	KF687762	KF687869	-
*C. rhexiae*	Coll1414, CBS 133134 *	*Rhexia virginica*	USA	JX145128	-	-	-	JX145179	JX145290
*C. salsolae*	ICMP 19051 *	*Salsola tragus*	Hungary	JX010242	JX009916	JX009562	JX009863	JX010403	KC888925
*C. serranegrense*	COAD 2100 *	*Cattleya jongheana*	Brazil	KY400111	-	KY407892	KY407894	KY407896	-
*C. siamense*	ICMP 18578, CBS 130417 *	*Coffea arabica*	Thailand	JX010171	JX009924	FJ907423	JX009865	JX010404	JQ899289
	ICMP 19118, CBS 130420	*Jasminum sambac*	Vietnam	HM131511	HM131497	HM131507	JX009895	JX010415	JQ807841
	ICMP 18642, CBS 125378	*Hymenocallis americana*	China	JX010278	JX010019	GQ856775	GQ856730	JX010410	JQ807842
	CBS 133239, GZAAS5.09506	*Murraya* sp.	China	JQ247633	JQ247609	JQ247657	-	JQ247644	KP703769
	CBS 133251, coll131, BPI 884113	*Vaccinium macrocarpon*	USA, New Jersey	JX145144	KP703275	-	-	JX145195	JX145313
	CBS 113199. CPC 2290	*Protea cynaroides*	Zimbabwe	KC297066	KC297008	KC296930	KC296985	KC297090	KP703763
	LC0149, PE007-2 (h)	*Camellia* sp.	China	KJ955079	KJ954780	KJ954361	-	KJ955228	KJ954495
	LF182	*Camellia* sp.	China	KJ955093	KJ954794	KJ954375	-	KJ955242	KJ954509
	CMM 3814	*Mangifera indica*	Brazil	KC702994	KC702955	KC702922	KC598113	KM404170	KJ155453
	**DL11**	** *Litchi chinensis* **	**China**	**OR461241**	**OR455977**	**OR456053**	**OR456015**	**OR456091**	**OR456119**
	**DL14**	** *Litchi chinensis* **	**China**	**OR461242**	**OR455978**	**OR456054**	**OR456016**	**OR456092**	**OR456120**
	**DL16**	** *Litchi chinensis* **	**China**	**OR461243**	**OR455979**	**OR456055**	**OR456017**	**OR456093**	**OR456121**
	**DL22**	** *Litchi chinensis* **	**China**	**OR461244**	**OR455980**	**OR456056**	**OR456018**	**OR456094**	**OR456122**
	**DL28**	** *Litchi chinensis* **	**China**	**OR461245**	**OR455981**	**OR456057**	**OR456019**	**OR456095**	**OR456123**
	**DL33**	** *Litchi chinensis* **	**China**	**OR461246**	**OR455982**	**OR456058**	**OR456020**	**OR456096**	**OR456124**
	**DL34**	** *Litchi chinensis* **	**China**	**OR461247**	**OR455983**	**OR456059**	**OR456021**	**OR456097**	**OR456125**
	**DL37**	** *Litchi chinensis* **	**China**	**OR461248**	**OR455984**	**OR456060**	**OR456022**	**OR456098**	**OR456126**
	**DL41**	** *Litchi chinensis* **	**China**	**OR461249**	**OR455985**	**OR456061**	**OR456023**	**OR456099**	**OR456127**
	**DL43**	** *Litchi chinensis* **	**China**	**OR461250**	**OR455986**	**OR456062**	**OR456024**	**OR456100**	**OR456128**
	**DL50**	** *Litchi chinensis* **	**China**	**OR461251**	**OR455987**	**OR456063**	**OR456025**	**OR456101**	**OR456129**
	**DL51**	** *Litchi chinensis* **	**China**	**OR461252**	**OR455988**	**OR456064**	**OR456026**	**OR456102**	**OR456130**
	**DL57**	** *Litchi chinensis* **	**China**	**OR461253**	**OR455989**	**OR456065**	**OR456027**	**OR456103**	**OR456131**
	**DL65**	** *Litchi chinensis* **	**China**	**OR461254**	**OR455990**	**OR456066**	**OR456028**	**OR456104**	**OR456132**
	**DL71**	** *Litchi chinensis* **	**China**	**OR461255**	**OR455991**	**OR456067**	**OR456029**	**OR456105**	**OR456133**
	**DL75**	** *Litchi chinensis* **	**China**	**OR461256**	**OR455992**	**OR456068**	**OR456030**	**OR456106**	**OR456134**
	**DL77**	** *Litchi chinensis* **	**China**	**OR461257**	**OR455993**	**OR456069**	**OR456031**	**OR456107**	**OR456135**
	**DL88**	** *Litchi chinensis* **	**China**	**OR461258**	**OR455994**	**OR456070**	**OR456032**	**OR456108**	**OR456136**
	**DL93**	** *Litchi chinensis* **	**China**	**OR461259**	**OR455995**	**OR456071**	**OR456033**	**OR456109**	**OR456137**
	**DL103**	** *Litchi chinensis* **	**China**	**OR461260**	**OR455996**	**OR456072**	**OR456034**	**OR456110**	**OR456138**
	**DL110**	** *Litchi chinensis* **	**China**	**OR461261**	**OR455997**	**OR456073**	**OR456035**	**OR456111**	**OR456139**
	**DL112**	** *Litchi chinensis* **	**China**	**OR461262**	**OR455998**	**OR456074**	**OR456036**	**OR456112**	**OR456140**
*C. sojae*	ATCC 62257 *	*Glycine max*	USA	MG600749	MG600810	MG600954	MG600860	MG601016	-
*C. subvariabile*	LC13876, NN040649 *	Unknown plant	China	MZ595883	MZ664054	MZ799343	MZ664181	MZ674001	-
*C. syzygicola*	MFLUCC10–0624 *	*Syzygium samarangense*	Thailand	KF242094	KF242156	KF157801	-	KF254880	-
	MFLUCC 10-0652	*Syzygium samarangense*	Thailand	KF242096	KF242158	KF157803	-	KF254882	-
	OCAC20	*Elettaria cardamomum*	India	KJ813596	KJ813546	KJ813446	KJ813496	KJ813471	KP743474
*C. temperatum*	Coll883, CBS133122 *	*Vaccinium macrocarpon*	USA	JX145159	-	-	-	JX145211	JX145298
*C. theobromicola*	ICMP 18649, CBS 124945 *	*Theobroma cacao*	Panama	JX010294	JX010006	JX009444	JX009869	JX010447	KC790726
*C. ti*	ICMP 4832 *	*Cordyline* sp.	New Zealand	JX010269	JX009952	JX009520	JX009898	JX010442	KM360146
*C. torulosum*	CBS 128544, ICMP 18586 *	*Solanum melongena*	New Zealand	JQ005164	JQ005251	JQ005512	JQ005338	JQ005598	-
*C. tropicale*	ICMP 18653, CBS 124949 *	*Theobroma cacao*	Panama	JX010264	JX010007	JX009489	JX009870	JX010407	KC790728
*C. variabile*	LC13875 *	Unknown plant	China	MZ595884	MZ664055	MZ799344	MZ664182	MZ674002	-
*C. vietnamense*	CBS 125477, BMT25(L3)	*Coffea* sp.	Vietnam	KF687720	KF687831	KF687791	KF687768	KF687876	-
	CBS 125478, Ld16(L2) *	*Coffea* sp.	Vietnam	KF687721	KF687832	KF687792	KF687769	KF687877	-
*C. viniferum*	GZAAS5.08601 *	*Vitis vinifera*, cv. ‘Shuijing’	China	JN412804	JN412798	JN412795	-	JN412813	-
*C. vittalense*	GUFCC 15503	*Calamus thwaitesii*	India	JN390935	KC790759	KC790646	KF451996	KC790892	-
	CBS 181.82 *	*Theobroma cacao*	India	MG600734	MG600796	MG600940	MG600851	MG601001	-
*C. wuxiense*	CGMCC 3.17894 *	*Camellia sinensis*	China	KU251591	KU252045	KU251672	KU251939	KU252200	KU251722
*C. xanthorrhoeae*	ICMP 17903, BRIP 45094, CBS 127831 *	*Xanthorrhoea preissii*	Australia	JX010261	JX009927	KC790635	JX009823	KC790913	KC790689
*C. zhaoqingense*	NN058985 *	On dead petiole of palm	China	MZ595905	MZ664065	MZ799304	MZ664203	MZ674023	-
	NN071035	On dead petiole of palm	China	MZ595906	MZ664066	MZ799305	MZ664204	MZ674024	-

* Ex-type culture. Strains studied in this paper are in **bold**.

**Table 2 jof-09-01042-t002:** Colony characteristics, sizes of conidia and growth rate of *Colletotrichum* species associated with anthracnose of litchi in this study.

Species	Colony Characteristics	Conidia					Growth Rates (mm/d)
		Shape	Length (μm)	Width (μm)	Means of Conidia Size	L/W Ratio	
*C. arecicola* (DL9)	White with black in center colony, dense mycelium	Cylindrical to clavate	13.1–17.4	4.5–6.0	15.1 × 5.1	2.9	11.6 ± 0.9
*C. fructicola* (DL44)	White with gray to black in the center, with orange conidial masses, dense mycelium	Cylindrical	10.4–18.4	4.3–5.9	14.7 × 5.2	2.8	11.9 ± 0.4
*C. gigasporum* (DL30)	Gray to pale green with a white margin, dense mycelium	Cylindrical	20.0–30.9	7.1–8.6	27.2 × 7.7	3.5	11.7 ± 0.1
*C. karstii* (DL64)	White colony, sparse mycelium	Cylindrical	13.0–18.0	5.7–7.5	15.4 × 6.7	2.3	7.6 ± 0.6
*C. musicola* (DL87)	Gray with a white margin, dense mycelium	Cylindrical to ellipsoidal	13.5–17.8	5.1–6.8	15.7 × 6.0	2.6	8.3 ± 0.5
*C. plurivorum* (DL62)	White to gray, dense mycelium	Cylindrical	14.8–19.5	4.8–7.7	16.9 × 6.2	2.7	12.0 ± 0.2
*C. siamense* (DL14)	White with gray in the center, with orange conidial masses, dense mycelium	Cylindrical	10.7–19.4	4.5–6.0	15.7 × 5.1	3.1	12.1 ± 0.1
*C. danzhouense* (DL52)	Gray to pale green with a white margin, dense mycelium	Cylindrical	14.4–21.6	5.6–7.2	17.6 × 6.5	2.7	7.4 ± 0.4

## Data Availability

Alignments generated during the current study are available from TreeBASE (http://treebase.org/treebase-web/home.html; study 30748, accessed on 6 September 2023). All sequence data are available in the NCBI GenBank, following the accession numbers in the manuscript.

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
