# Peer review of "Characterization of *Colletotrichum* Species Infecting Litchi in Hainan, China"

_jof, 2023, doi:10.3390/jof9111042_

Round 1

Reviewer 1 Report

This article provides an account of species of Colletotrichum on Litchi that cause anthracnose. One new species is described. In general the article is adequate using a multigene phylogeny to determine the identification of species and relationships among species complexes and species.

Some editorial issues are present throughout the paper as follows:

When proper names are presented such as species names, they should always be listed in alphabetical order unless there is some other logical order for this. It should not be random!

The authors should decide how to use the names for species complexes e.g. gloeosporioides complex or C. gloeosporioides complex. I suggest using the latter. At present there is a mixture of formats in use.

Between a range of numbers or items such as figs. 1–4 or a–c, an en dash should be used, as found in Word under Symbols, more symbols, special characters, rather than a hyphen. I found an en dash occasionally in use but usually not. Please change all of these.

The numbers one through ten should be written out while, for numbers above ten, the numeral e.g. 79 can be used.

When China is used as an adjective, it should be surrounded by commas as needed on p. 1, ln 21; p. 2, ln 47.

The word mycelia should not be used as it is the same as mycelium. Use mycelium. This is the word for a collection of hyphae. At present both are used.

Additional editorial comments:

p. 1, ln 16, in Abstract, the first Litchi should not be in bold.

ln 22, delete the comma

lns 24, 25, 32, and keywords, these names should be listed alphabetically. This is true throughout the paper and will not be mentioned again.  Please change this throughout the paper.

p. 2, ln 7, add a comma before which. This is true throughout the paper.

ln 21, delete “was”

ln 27, change “a lot of” (slang) to “many”

ln 40, these reports indicate that

ln 41, change “varies” to “vary”

p. 3, lns. 35-37, this sentence needs a verb

ln 16, 19, 20, the punctuation here is unclear. Put an “and” before the last phrase.

p. 6, under C. cordylinicola, Cordyline is misspelled

p. 7, under C. musicola, Colocasia should start with a capital letter

p. 8, under C. phyllanthi, the host should be in italics

under C. proteae, delete the sp. after Proteaceae as not needed

under C. salsolae, what is the country?

Under C. siamense, the hosts should be in italics.

p. 9, under C. temperatum, there should not be a comma after USA

p. 10, under C. ti, the generic name Cordyline should be in italics.

p. 11, ln 4, Mycelial should not be in caps

lns 18-19, delete the commas

ln 35, 36, Multilocus should not be hyphenated.

p. 12, Fig. 1, in the caption and for other captions, I suggest mentioning that the isolates in bold from the study were isolates from litchi.

p. 13, ln 70, close the space before the =

In Figure 2, label isolate DL52 and DL106 with the name of the new species

p. 14, ln 88 and 89, What is Danzhou? Is it a city or something else?

ln 901, put a comma after the Holotype information.

ln 100, what are the growth rates on the culture on OA and SNA?

Lns 90, 105, take out the period after May. That is not an abbreviation so there is no need for a period.

Ln 113, respectively is misspelled

p. 15, illustration is not aligned with the text

p. 16, ln 133, delete “All the…”

p. 17, illustration is not aligned with the text

ln 157, I do not understand the second half of this sentence. This starts with “and three of them…” This could be divided into two sentences.

Ln 170 vs. 174, do you want to say “broad host range” or “large host range”. I prefer “broad” just be consistent.

p. 18, ln 181, delete “was”

ln 187, 188, where was this species reported?

Ln 192, delete “then for…”

Ln 210, change “first” to “for the first time”

The references were not checked.

Fairly good except for the misspelling and language problems mentioned abov.

Author Response

Dear Reviewer,

We would like to thank you for the valuable comments which have enabled us to improve the manuscript entitled “Characterization of Colletotrichum species infecting litchi in Hainan, China” (jof-2655301). We addressed all comments and modified the manuscript accordingly. The major changes are highlighted in yellow in the manuscript and our responses to your comments are listed below in italics.

 When proper names are presented such as species names, they should always be listed in alphabetical order unless there is some other logical order for this. It should not be random!

Yes, the order was changed to be in alphabetical order

The authors should decide how to use the names for species complexes e.g. gloeosporioides complex or C. gloeosporioides complex. I suggest using the latter. At present there is a mixture of formats in use.

We modified the text to use the latter format for the names of species complexes.

Between a range of numbers or items such as figs. 1–4 or a–c, an en dash should be used, as found in Word under Symbols, more symbols, special characters, rather than a hyphen. I found an en dash occasionally in use but usually not. Please change all of these.

This has been changed.

The numbers one through ten should be written out while, for numbers above ten, the numeral e.g. 79 can be used.

We changed this according to the comment.

When China is used as an adjective, it should be surrounded by commas as needed on p. 1, ln 21; p. 2, ln 47.

Changed.

The word mycelia should not be used as it is the same as mycelium. Use mycelium. This is the word for a collection of hyphae. At present both are used.

Yes, only ‘mycelium’ is now used in the paper.

Additional editorial comments:

  1. 1, ln 16, in Abstract, the first Litchi should not be in bold.

Changed.

ln 22, delete the comma

Deleted.

lns 24, 25, 32, and keywords, these names should be listed alphabetically. This is true throughout the paper and will not be mentioned again.  Please change this throughout the paper.

Yes, these names were listed alphabetically and we changed this throughout the paper.

  1. 2, ln 7, add a comma before which. This is true throughout the paper.

Yes, a comma was added before which and we changed this throughout the paper.

ln 21, delete “was”

Deleted

ln 27, change “a lot of” (slang) to “many”

Changed.

ln 40, these reports indicate that

Changed.

ln 41, change “varies” to “vary”

Changed.

  1. 3, lns. 35-37, this sentence needs a verb

A verb was added.

ln 16, 19, 20, the punctuation here is unclear. Put an “and” before the last phrase.

Revised.

  1. 6, under C. cordylinicola, Cordyline is misspelled

Revised.

  1. 7, under C. musicola, Colocasia should start with a capital letter

Revised.

  1. 8, under C. phyllanthi, the host should be in italics

Revised.

under C. proteae, delete the sp. after Proteaceae as not needed

Deleted.

under C. salsolae, what is the country?

The country (Hungary) was added.

Under C. siamense, the hosts should be in italics.

Revised.

  1. 9, under C. temperatum, there should not be a comma after USA

The comma was deleted.

  1. 10, under C. ti, the generic name Cordyline should be in italics.

Revised.

  1. 11, ln 4, Mycelial should not be in caps

Revised.

lns 18-19, delete the commas

Deleted.

ln 35, 36, Multilocus should not be hyphenated.

Revised.

  1. 12, Fig. 1, in the caption and for other captions, I suggest mentioning that the isolates in bold from the study were isolates from litchi.

Colored blocks indicate clades including isolates obtained from litchi in this study. Ex-type isolates were shown in bold.

  1. 13, ln 70, close the space before the =

Revised.

In Figure 2, label isolate DL52 and DL106 with the name of the new species

Revised.

  1. 14, ln 88 and 89, What is Danzhou? Is it a city or something else?

Yes, it is a city and we added this.

ln 90, put a comma after the Holotype information.

A comma was added here.

ln 100, what are the growth rates on the culture on OA and SNA?

We added this information here.

Lns 90, 105, take out the period after May. That is not an abbreviation so there is no need for a period.

Revised.

Ln 113, respectively is misspelled

Revised.

  1. 15, illustration is not aligned with the text

The technical editor put the figure and table here

  1. 16, ln 133, delete “All the…”

Deleted.

  1. 17, illustration is not aligned with the text

The technical editor put the figures here.

ln 157, I do not understand the second half of this sentence. This starts with “and three of them…” This could be divided into two sentences.

Revised.

Ln 170 vs. 174, do you want to say “broad host range” or “large host range”. I prefer “broad” just be consistent.

Yes, broad was used here.

  1. 18, ln 181, delete “was”

Deleted.

ln 187, 188, where was this species reported?

Added here.

Ln 192, delete “then for…”

Deleted.

Ln 210, change “first” to “for the first time”

Changed.

Reviewer 2 Report

Overall it is well done.  However, I assume you did use BLAST with the individual ITS etc sequences etc. that you developed.  Especially since you report a new species, it would be good to state that 'all' were most similar to Colletotrichum sp, assuming that is correct.   

I have attached an edited version of the reviewed ms for the authors use

Very minor, my suggestions are shown on the edited pdf returned to the editor

Author Response

Dear Reviewer,

We would like to thank you for the valuable comments which have enabled us to improve the manuscript entitled “Characterization of Colletotrichum species infecting litchi in Hainan, China” (jof-2655301). We addressed all comments and modified the manuscript accordingly. The major changes are highlighted in yellow in the text and our responses to your comments are listed below in italics.

Overall it is well done.  However, I assume you did use BLAST with the individual ITS etc sequences etc. that you developed.  Especially since you report a new species, it would be good to state that 'all' were most similar to Colletotrichum sp, assuming that is correct.  

Yes, we added this.

  1. 1, lns. 22-23 change to identified that eight Colletotrichum species, from four species complexes

Changed.

  1. 2, lns. 21 delete was

Deleted.

  1. 2, lns. 24 change to Multilocus

Changed.

  1. 2, lns. 27 change to many new Colletotrichum species have been

Changed.

  1. 3, lns. 50 change to Consensus sequences were obtained by assembling

Changed.

  1. 4, lns. 32 delete was set

Deleted.

  1. 11, lns. 5 change on to to

Changed.

  1. 17, lns. 162 change were to are

Changed.

Reviewer 3 Report

Dear Authors,

I think that your manuscript is adequately conceived and organized in every section opening to a clear and actual framework of interesting topic. Given the increasing crop importance in China and worlwide, I also think the paper’s topic is very useful and it fully accomplishes the scope of JoF.

Moreover, the quality of the paper is very high and my impression is positive. Comprehensively, the paper include new and very interesting information that could be very useful for the researchers and technicians and it should be published.

However many modifications and comments should be performed and/or addressed prior to publication. I provide in attachment an annotated PDF. Please fulfill all included requirements.

Author Response

Dear Reviewer,

We would like to thank you for the valuable comments which have enabled us to improve the manuscript entitled “Characterization of Colletotrichum species infecting litchi in Hainan, China” (jof-2655301). We addressed all comments and modified the manuscript accordingly. The major changes are highlighted in yellow in the text and our response to reviewers is in italics.

However many modifications and comments should be performed and/or addressed prior to publication. I provide in attachment an annotated PDF. Please fulfill all included requirements.

We thank the reviewer for providing edits in the PDF. We can confirm that the required edits in the PDF were changed and supplemented accordingly.

  1. 1, ln 16, Please, not in bold.

Changed.

  1. 1, lns 24, 25, 32 Please, the authors should report the entire name of species complexes (e.g. C. gloeosporioides species complex). Do the same for all manuscript

Yes, the entire name of species complexes was reported and we changed this throughout the manuscript.

  1. 5 This table is very long but I understand the information are necessary. However, the authors could group some similar information (e.g. C. arecicola from Areca catechu in China; C. musicola from Musa in China; all C, fructicola from Litchi in China; C. siamense from Litchi in China and so on to synthetize as soon as possible the table)

The species names were listed in alphabetical order as one of the Reviewers suggested.

  1. 11 Please, in this section experimental scheme data are missing. How many replicates? how many seedlings for each replicates? Since the authors performed twice the pathogenicity proof, have the authors performed statistical analysis to detecte significant differences or interactions (inoculation method x trial)?

Yes, the experimental scheme data was added. Significant differences were detected among species, which is now indicated.

  1. 12 Please, enlarge and made it longer to improve readability. The resolution should be improved too.

A photo with higher resolution was added here.

  1. 13 Please, enlarge and made it longer to improve readability. The resolution should be improved too.

A photo with higher resolution was added here.

  1. 17 The authors could think to a possible data analysis to verify there are significant differences in severity among tested Colletotrichum species

Yes, the differences in severity among tested Colletotrichum species were analysed statistically and were found to be significant. This was added..